

# Evaluation of the association of physical activity levels with self-perceived health, depression, and anxiety in Spanish individuals with high cholesterol levels: a retrospective cross-sectional study

Ángel Denche-Zamorano[1], Jofre Pisà-Canyelles[2], Sabina Barrios-Fernández[3], Antonio Castillo-Paredes[4], Raquel Pastor-Cisneros[1], Maria Mendoza-Muñoz[5,6], Diana Salas Gómez[7] and Cristina Mendoza Holgado[3]

[1] Promoting a Healthy Society Research Group (PHeSO), University of Extremadura, Cáceres, Spain
[2] Health, Economy, Motricity and Education Research Group (HEME), University of Extremadura, Cáceres, Spain
[3] Occupation, Participation, Sustainability and Quality of Life (Ability Research Group), University of Extremadura, Cáceres, Spain
[4] Grupo AFySE, Investigación en Actividad Física y Salud Escolar, Escuela de Pedagogía en Educación Física, Facultad de Educación, Universidad de Las Americas, Santiago, Chile
[5] Research Group on Physical and Health Literacy and Health-Related Quality of Life (PHYQOL), University of Extremadura, Cáceres, Spain
[6] Departamento de Desporto e Saúde, Escola de Saúde e Desenvolvimento Humano, Universidade de Evora, Évora, Portugal
[7] Escuelas Universitarias Gimbernat (EUG), Physiotherapy School Cantabria, Movement Analysis Laboratory, Universidad de Cantabria, Torrelavega, Spain

Corresponding author
Antonio Castillo-Paredes, acastillop85@gmail.com

## ABSTRACT

**Background**. Hypercholesterolemia is the most common form of dyslipidaemia in the world leading to negative health effects, both physical and mental. Physical activity (PA) can reduce total cholesterol and has positive effects on mental health. This retrospective cross-sectional study analyses the relationships between physical activity level (PAL), self-perceived health (SPH) and mental health.

**Methods**. This study was based on data from the Spanish National Health Survey 2017 (SNHS 2017), with 3,176 Spanish adults with high cholesterol as participants. Non-parametric tests were used as the data did not follow normality.

**Results**. Dependent relationships were found between PAL and SPH, depression and anxiety. Women had higher depression and anxiety prevalences than men, while men were more likely to report being very active, although the proportion of walkers was higher in women. The physically inactive population presented higher negative SPH, depression and anxiety proportions and psychological distress than physically active people.

**Conclusion**. The physically inactive people had a higher risk of negative SPH, depression and anxiety. Regular PA may improve SPH and mental health in people with high cholesterol, but more studies are needed to establish causal relationships, mechanisms, and optimal doses.

## INTRODUCTION

Elevated blood cholesterol, hypercholesterolemia (*Adeloye et al., 2020*), is the most common form of dyslipidaemia in the world (*Pirillo et al., 2021*). It has a high worldwide prevalence, estimated a 54% in the European population and 48% in the American (*Timmis et al., 2018*). It causes adverse health effects related to 33% of coronary heart disease cases worldwide (*Pirillo et al., 2021*). It is one of the main risk factors for cardiovascular, cerebrovascular and peripheral vascular diseases (*Sharrett et al., 2001*; *Curb et al., 2004*; *Tunstall-Pedoe et al., 1994*; *Al-Zahrani et al., 2021*), all major diseases with high mortality and morbidity in industrialized countries (*Grau et al., 2007*; *Ivanovic & Tadic, 2015*). Moreover, high cholesterol is linked to many chronic diseases such as high blood pressure (hypertension) (*Martone et al., 2022*), diabetes (*Tajima et al., 2014*), obesity (*Al-Zahrani et al., 2021*), and even erectile dysfunction in men (*Musicki et al., 2010*). Globally, adult suffer from diseases and health problems which, among other factors, are due to high cholesterol levels as a consequence of unhealthy lifestyles (*Nogueira de Sá et al., 2022*). Consequently, hypercholesterolemia management is one of the major global public health concerns. It is a priority to develop prevention strategies to better control one of the most important modifiable risk factors of cardiovascular disease (*Martone et al., 2022*). Among the prevention strategies that could help control hypercholesterolemia, it is also essential to take into account other factors associated with it that are documented in the literature, such as sociodemographic characteristics, an inadequate lifestyle, alterations in body mass index (BMI) or low self-assessed health status (*Nogueira de Sá et al., 2022*).

### Theorical framework: cholesterol and mental disorders

Although less discussed in the literature, total cholesterol levels (Includes both types: low-density lipoprotein (LDL) cholesterol and high-density lipoprotein (HDL) cholesterol) are also associated with mental disorders or mental health problems (*Pereira, 2017*; *Troisi, 2009*). Anxiety and depression are the most prevalent mental disorders in the general population (*Kessler et al., 2009*). The study by *Han (2022)* reports an odd ratio of presenting depressive symptoms of 2.28 (1.07–4.86) in the high cholesterol group with respect to the group with normal cholesterol values. In addition, in people without psychiatric pathology, an association has been found between elevated cholesterol levels and symptoms of anxiety or mild depressive states (*Rafter, 2001*; *Cheon, 2023*). In the same way, high cholesterol levels have also been reported in female patients with panic disorders (*Bajwa et al., 1992*; *You et al., 2023*). In contrast, some studies suggest that naturally low concentrations of lipids and lipoproteins are associated with depression as well as poor psychological health (*Rafter, 2001*; *Suarez, 1999*). One hypothesis would be that low cholesterol levels influence depression as they may reduce the availability of serotonin, making the patient more susceptible to depression (*Rafter, 2001*; *Papakostas et al., 2004*). In this respect, a recent meta-analysis revealed that elevated triglyceride levels and reduced HDL levels

were associated with the first episode of a major depressive disorder (*Wei et al., 2020*). Contrary to these findings, other studies report that LDL levels is also associated with depression (*Tedders et al., 2011*; *Andruškiene et al., 2013*). Regarding the inconsistencies of the results in the literature between cholesterol levels and depression, it could be that a modifying factor could be sex, making the relationship between cholesterol and depression different in men and women, but it is not yet clear (*Tedders et al., 2011*). For example, a study performed with an American population found that low HDL levels were associated with higher levels of major depression in women but not in men, while in men, low and high LDL levels and total cholesterol were associated with major depression (*Tedders et al., 2011*). Although there is evidence of an association between cholesterol and psychological symptoms, a causal relationship cannot be established as cholesterol itself does not directly cause behaviours, it may induce chemical changes affecting the likelihood of certain behavioural outcomes by modulating certain neural pathways (*Pereira, 2017*). As commented previously, literature also shows that self-rated health status is associated with dyslipidaemia (*Nogueira de Sá et al., 2022*). In this sense, a study in a Brazilian population found an association between high cholesterol levels and poorer self-rated health status. It is important to consider that self-rated health status as well as symptoms related to anxiety or depression when caring for a person because they are important predictors of mortality and morbidity (*Nogueira de Sá et al., 2022*; *Larsen et al., 2014*).

## Physical activity, mental health and cholesterol

On the other hand, another important factor associated with dyslipidaemia is lifestyle. Literature establishes the importance of regular physical activity to prevent and control dyslipidaemia (*Tian & Meng, 2019*). In addition, evidence suggests that sedentary or physically inactive behaviour is associated with mental disorders such as depression and anxiety in general population (*Harris, 2018*). In this sense, several studies reported an association of PA with cholesterol levels, for example, higher levels of PA are associated with higher HDL levels (*Kokkinos & Fernhall, 1999*). Moreover to the apparent physical health benefits, PA also positively affects mental health, being associated with reduced depression and anxiety symptoms and improved cognitive functioning and psychological well-being (*Carek, Laibstain & Carek, 2011*). A previous study carried out our investigator group found associations between mental health and PA levels in the Spanish population in the pre-pandemic period analysing the data of the Spanish National Health Survey (SNHS 2017) (*Denche-Zamorano et al., 2022a*). Specifically, Spanish general population groups with higher level of PA had fewer mental disorders evaluate with the Goldberg General Health Questionnaire (GHQ-12) (*Denche-Zamorano et al., 2022a*). The National Health Survey (NHS) is carried out in Spain by the Ministry of Health, Consumer Affairs and Social Welfare (MSCBS) every 5 years, so the SNHS 2017 is the last health survey before pandemic period. The GHQ-12 has been used in the SNHS to analyse the evolution of the population's mental health or check the impact of specific situations, such as economic crises or pandemics (*Cabrera-León et al., 2017*). In other way, a previous study has also reported that mental illness was associated with fewer walking days in a Spanish population with

 

a common disease such as asthma (*De-Miguel-Diez et al., 2024*). In addition, other study show that patients with asthma who practice PA have less psychological distress (*Denche-Zamorano et al., 2022e*). However, to the authors' knowledge there are no data on the association between mental disorders such as depression and anxiety and physical activity levels in a sample of Spanish population with high cholesterol also in the pre-pandemic period in order to be able to compare the impact of the pandemic in the future. The author considerate that improving physical fitness could be a strategy to address the impact of an unhealthy lifestyle on depression and anxiety in Spanish people with high cholesterol (*Alam & Rufo, 2019*).

According to the literature cited above, the association between cholesterol and depressive or anxiety disorders is complex, probably runs in one direction or the other, the underlying mechanism remains unknown, and causality has not been established. This suggests that more studies are needed to investigate and deepen the links between these variables and how are modulated for other factors like PA.

The results of this study may inform and help health promotion programs to develop more specific protocols aimed at targeting people with hypercholesterolemia as well as their possible related symptoms such as depression or anxiety through physical activity. In addition, the results will provide baseline data on mental health in this population before the COVID-19 pandemic, which will allow the establishment of lines of research to evaluate the effects of the pandemic on mental health in people with high cholesterol.

The current study aimed to analyse associations and differences between different PA levels and mental health, self-perceived health, anxiety, depression and sex in Spanish adults with high cholesterol were assessed. In addition, predictors and risk factors for mental health, anxiety and depression.

## MATERIALS AND METHODS

### Study design and sample

The data used in this study is a second analysis of data from the SNHS 2017 and can be downloaded freely and without restriction of: website of the Spanish Ministry of Health (*Ministry of Health, 2017a*). A retrospective cross-sectional study was conducted with an adult household resident population (over 15 years of age) living in Spain. in collaboration with the National Institute of Statistics (NIS), collecting health information on the population residing in Spain to ascertain their state of health. For this purpose, the adult questionnaire was used in the SNHS 2017, covering four broad areas: socio-demographic variables, health status, use of health services and health determinants (*Ministry of Health, 2017a*). A stratified three-phase random sampling system was used to obtain the sample for the SNHS 2017. The surveys were conducted between October 2016 and October 2017. The SNHS 2017 methodology includes all the information on the regulations governing the survey, the objectives, the design of the questionnaire, the sample design (type of sampling, sample size, sample selection, distribution over time, estimators and sampling errors), information collection, treatment of erroneous or missing data, among other important information (*Ministerio de Sanidad—Sanidad En Datos—Encuesta Nacional de Salud de España, 2017b*).

The SNHS 2017 sample size was 23,089 participants, being the initial sample of 23,089 adults, aged 15 years and older, living in family dwellings in Spain. The following selection criteria were applied to achieve the final sample for current study: being under 70 years of age, reporting high cholesterol, submitting data on depression and anxiety, submitting data on mental health variables, and submitting data on PA variables. Public files with anonymized SNHS 2017 microdata were downloaded.

In order to select participants with high cholesterol, information was extracted from the following survey questions:

(a) Do you have, or have you ever had, high cholesterol? (Yes, No, Don't know or No answer (DK/NA)).
(b) Have you suffered from it in the last 12 months? (Yes, No, DK/NA).
(c) Has a doctor told you that you have it? (Yes, No, DK/NA).

To be considered as having high cholesterol, a participant had to have answered "Yes" to (a) and "Yes" to (b) or (c). A detailed sample selection procedure is explained in Fig. 1.

## Procedures and variables

Data were extracted from the following questions:

### Outcomes variables
### Self-perceived health

(a) In the last 12 months, would you say your health status has been very good, good, fair, fair, bad, or very bad?. These responses were grouped into: SPH Positive (Good and Very Good), SPH Fair (Fair) and Negative (Bad and Very Bad).

### Depression status

(a) Do you suffer or have you ever suffered from depression? (Yes, No, DK/NA).
(b) Have you suffered from it in the last 12 months? (Yes, No, DK/NA).
(c) Has a doctor told you that you have it? (Yes, No, DK/NA).

To be considered as having depression, a participant had to have answered "Yes" to (a) and "Yes" to (b) or (c).

### Chronic anxiety status

(a) Do you suffer, or have you ever suffered from chronic anxiety? (Yes, No, DK/NA).
(b) Have you suffered from it in the last 12 months? (Yes, No, DK/NA).
(c) Has a doctor told you that you have it? (Yes, No, DK/NA).

To be considered as having high cholesterol, a participant had to have answered "Yes" to (a) and "Yes" to (b) or (c).

### Mental health

Constructed from the sum of participants' responses to item corresponding to the Goldberg General Health Questionnaire (GHQ-12) (*Goldberg & Williams, 1988*; *Muñoz Bermejo et al., 2020*). This questionnaire measures the current mental health of the respondent, through the answers given to 12 items that can take values from 0 to 3. Mental health is evaluated, according to the global score of the questionnaire, with values between 0 and 36 points, where 0 is the best mental health and 36, is the worst. A previous study presented

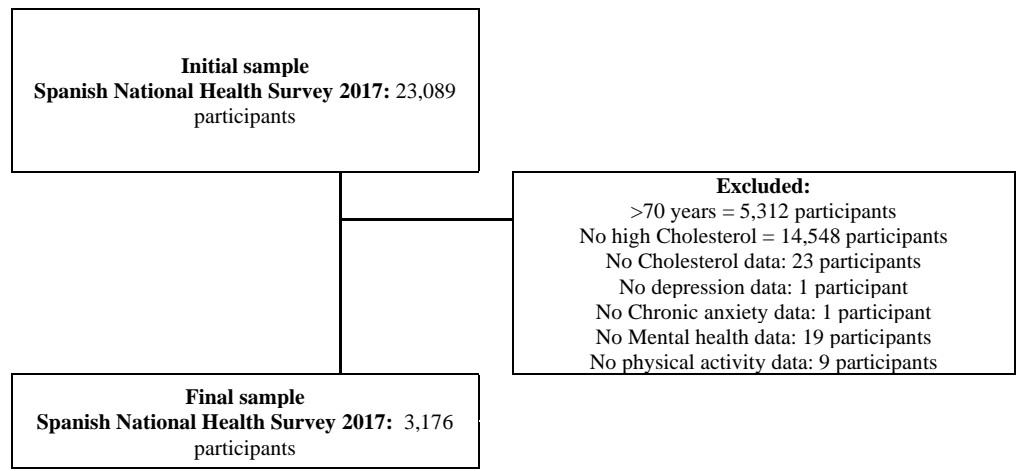

**Figure 1   Flowchart outlining the study sample's eligibility criteria.**

the psychometric properties of this questionnaire and established that for the Spanish population, the respondent is considered to have an emotional disorder if his or her score is higher than 12 points (*Muñoz Bermejo et al., 2020*; *Sánchez-López & Dresch, 2008*).

### Mental health status

From the sum of the responses to GHQ-12, a dichotomous variable was also created.

 (a) Scores ≤12; No mental health disorders

 (b) Scores >12; Yes mental health disorders

### Exploratory variables
### Physical Activity Index

(a) It was generated from participants' responses to question that asked participants about the intensity, frequency and duration of physical activity performed in the last week. From the responses, a series of factors were applied to generate the Physical Activity Index (PAI), which could take values between 0 and 67.5. The PAI formula was described in previous publications (*Denche-Zamorano et al., 2022b*; *Nes et al., 2011*).

### Physical Activity Level

(a) Based on the PAI and the question: How often they walked more than 10 consecutive minutes per week.

 Participants were grouped into "Inactive", "Walkers", "Active" and "Very Active". Previous research has defined how these groupings were made (*Denche-Zamorano et al., 2022b*).

 Age (in years).

 Sex (men or women).

### Social class

(a) This variable grouped participants into six social categories according to the proposal of the Spanish Society of Epidemiology's Working Group on Determinants where social

class is assigned according to occupation: I, II, III, IV, V and VI. I, managers of business establishments with 10 or more employees and professionals with university qualifications; II, managers of business establishments with fewer than 10 employees, professionals with university qualifications and other support technicians, sportsmen and artists; III, intermediate occupations and the freelancers; IV, supervisors and qualified technical workers; V, skilled workers in the primary sector and other semi-skilled workers; or VI, unqualified workers.

## Statistical analysis

The distribution followed by the data of the variables of interest was analysed using the Kolgomorov-Smirnov test. A descriptive analysis was performed to assess the gender differences in age and mental health (continuous variables) and the prevalence of SPH, depression status, anxiety status and physical activity level (PAL) (categorical variables). To evaluate sex differences in the variables of interest was using the Mann–Whitney U test for the continuous variables or the Chi-square test with a *post-hoc* test pairwise z-test for categorical variables with Bonferroni correction. In addition, the Chi-square test was used to evaluate the relationship between PAL and depression status, anxiety status and SPH; as well as the difference in proportions according to the PAL group. Then to interpret the strength of association, Phi coefficient (2x2 contingency tables) or Cramer's V Coefficient (contingency tables greater than 2x2) were calculated according to the following interpretation: insignificant (0.0–0.10), weak (0.10–0.20), moderate (0.20–0.40), relatively strong (0.40–0.60), strong (0.60–0.80), very strong (0.80–1.00) (*Lee, 2016*). To test mental health scores according to the PAL with the Kruskal-Wallis test to analyse possible differences in scores at baseline, and the Mann–Whitney U test to analyse differences between the different levels.

Multiple binary logistic regressions were performed, taking the depression status, anxiety status and mental health status respectively, as dependent variables, and sex, age, PAL and social class as independent variables. For all logistic binary regressions, the assumptions of the normality of residuals, the absence of influential factors were tested using Cook's distance and the absence of outliers. For all analyses, a significance level of less than 0.05 was assumed as statistically significant, and IBM SPSS Statistical v.25 (Armonk, NY, USA) was used.

# RESULTS

The final sample analyse was 3,176 participants, aged 15 years and older with high cholesterol, residing in Spanish family dwellings. The median age of the sample was 58 years (57 years in men and 59 years in women, $p < 0.001$). The median of the GHQ-12 scores was 10 points. Women presented higher scores of psychological distress, according to the GHQ-12 than men (11 *vs* 10, $p < 0.001$). Dependency relationships were found between sex and SPH ($X^2 = 19.4$, $df = 2$, $p < 0.001$, $V = 0.078$), prevalence of depression ($X^2 = 101.8$, $df = 2$, $p < 0.001$, $\Phi = 0.179$), and anxiety ($X^2 = 89.3$, $df = 2$, $p < 0.001$, $\Phi = 0.168$). A higher proportion of positive SPH was found in men than in women (58.5% *vs* 50.9%, $p < 0.05$). Women presented a higher prevalence of depression (24.8% *vs* 11.1%,

**Table 1** Descriptive analysis of the overall population: age, mental health, self-perceived health, depression and anxiety prevalences and physical activity level. Comparison between sexes and dependence analysis between categorical variables and sex.

| Variables Age | Men n = 1,623 | | Women n = 1,553 | | Total n = 3,176 | | $X^2$ | df | p-value Mann-Whithey U test | V |
|---|---|---|---|---|---|---|---|---|---|---|
| Median (IQR) | 57 | (15) | 59 | (13) | 58 | (14) | – | – | <0.001 | – |
| Mean (SD) | 55.4 | (10.0) | 57.0 | (9.9) | 56.2 | (10.0) | – | – | – | – |
| Mental health | | | | | | | | | | |
| Median (IQR) | 10 | (5) | 11 | (6) | 10 | (5) | – | – | <0.001 | – |
| Mean (SD) | 10.4 | (4.8) | 11.9 | (5.5) | 11.1 | (5.2) | – | – | – | – |
| SPH | n | % | n | % | n | % | $X^2$ | df | p-value Chi-square test | V |
| Negative | 188 | 11.6 | 230 | 14.8* | 418 | 13.2 | | | | |
| Fair | 485 | 29.9 | 532 | 34.3* | 1,017 | 32.0 | 19.4 | 2 | <0.001 | 0.078 |
| Positive | 950 | 58.5 | 791 | 50.9* | 1,741 | 54.8 | | | | |
| Depression | | | | | | | | | | Φ |
| Yes | 180 | 11.1 | 385 | 24.8* | 565 | 17.8 | 101.8 | 2 | <0.001 | 0.179 |
| No | 1,443 | 88.9 | 1,168 | 75.2* | 2,611 | 82.2 | | | | |
| Anxiety | | | | | | | | | | Φ |
| Yes | 172 | 10.6 | 359 | 23.1* | 531 | 16.7 | 89.3 | 2 | <0.001 | 0.168 |
| No | 1,451 | 89.4 | 1,194 | 76.9* | 2,649 | 83.3 | | | | |
| PAL | | | | | | | | | | V |
| Inactive | 240 | 14.8 | 241 | 15.5 | 481 | 15.1 | | | | |
| Walker | 813 | 50.1 | 865 | 55.7* | 1,678 | 52.8 | 28.0 | 3 | <0.001 | 0.094 |
| Active | 416 | 25.6 | 369 | 23.8 | 785 | 24.7 | | | | |
| Very active | 154 | 9.5 | 78 | 5.0* | 232 | 7.3 | | | | |

**Notes.**

$X^2$, Pearson's Chi-square; df, degrees of freedom; n, participants; %, percentage; IQR, Interquartile Range; SD, Standard Deviation; SPH, Self-perceived healh; –, Not applicable.

*Significant differences in proportions between men and women with p-value < 0.05 in z-test for independent proportions.

PAL, Physical Activity Level; V, Cramer's V coefficient; Φ, Phi coeficient.

$p < 0.05$) and anxiety (23.1% vs 10.6%, $p < 0.05$) than men (Table 1). Figure 2 shows the proportions of positive SPH, depression and anxiety of people with high cholesterol according to their sex. Also, dependency relationships were found between PAL and sex ($X^2 = 28.0$, $df = 3$, $p < 0.001$, $V = 0.094$). The proportion of men in "very active" group was found to be higher than that of women (9.5% vs 5.0%, $p < 0.05$). In contrast, the proportion of walking women was higher than that of men (55.7% vs 50.1%, $p < 0.05$) (Table 1).

## Relationships between PAL and depression, anxiety and SPH

Self-reported depression was found to be associated with PAL ($X^2 = 78.7$, $df = 3$, $p < 0.001$, $V = 0.157$), and the highest and the lowest prevalence of depression was found in "Inactive" and "Active" groups, respectively (28.1% vs 9.6%, $p < 0.05$) (Table 2). PAL and self-reported anxiety were also found to be associated ($X^2 = 30.6$, $df = 3$, $p < 0.001$, $V = 0.098$). The highest and lowest proportions of anxiety were found in the "Inactive" and "Active"
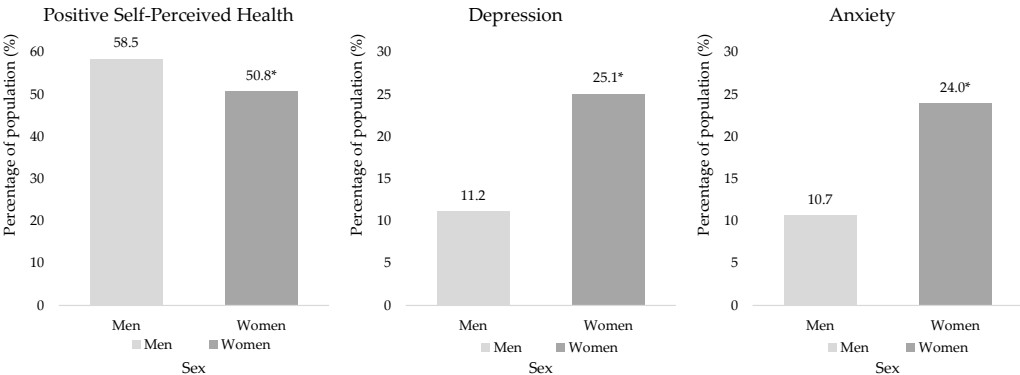

**Figure 2** **Proportions of positive SPH, depression, and anxiety in people with high cholesterol, according to sex.** An asterisk (*) indicates significant differences between sex ratios with $p < 0.05$ from pairwise $z$-test.

**Table 2** **Self-perceived health, prevalence of depression and anxiety in people with high cholesterol as a function of physical activity level. Intergroup comparisons and dependency analysis between depression, anxiety, and self-perceived health; and level of physical activity.**

| Variables | Inactive $n$ (%) | | Walker $n$ (%) | | Active $n$ (%) | | Very active $n$ (%) | | Total $n$ (%) | | $X^2$ | df | $p$ | V |
|---|---|---|---|---|---|---|---|---|---|---|---|---|---|---|
| **Depression** | | | | | | | | | | | | | | |
| Yes | 135[a] | (28.1) | 326[b] | (19.4) | 75[c] | (9.6) | 29[bc] | (12.5) | 565 | (17.8) | 78.7 | 3 | <0.001 | 0.157 |
| No | 346[a] | (71.9) | 1,353[b] | (80.6) | 710[c] | (90.4) | 203[bc] | (87.5) | 2,611 | (82.2) | | | | |
| **Anxiety** | | | | | | | | | | | | | | |
| Yes | 115[a] | (23.9) | 288[b] | (17.2) | 98[c] | (12.5) | 30[bc] | (12.9) | 531 | (16.7) | 30.6 | 3 | <0.001 | 0.098 |
| No | 366[a] | (76.1) | 1,390[b] | (82.8) | 687[c] | (87.5) | 202[bc] | (87.1) | 2,645 | (83.3) | | | | |
| **Self-perceived health** | | | | | | | | | | | | | | |
| Negative | 145[a] | (30.1) | 215[b] | (12.8) | 48[c] | (6.1) | 10[c] | (4.3) | 418 | (13.2) | | | | |
| Fair | 168[a] | (34.9) | 587[a] | (35.0) | 210[b] | (26.8) | 52[b] | (22.4) | 1,017 | (32.0) | 241.3 | 6 | <0.001 | 0.195 |
| Positive | 178[a] | (34.9) | 876[b] | (52.2) | 527[c] | (67.1) | 170[c] | (73.3) | 1,741 | (54.7) | | | | |

**Notes.**

$X^2$, Pearson's Chi-square; df, Degrees of freedom; $p$, $p$-value; $n$, participants; %, percentage).

[abc] Different subscripts indicate significant differences in proportions between physical activity groups with $p$-value $< 0.05$ in $z$-test for independent proportions.

V, Cramer's V coefficient.

groups, respectively (23.9% *vs* 12.5%, $p < 0.05$). Similarly, dependency relationships were found between PAL and SPH ($X^2 = 241.3$, $df = 6$, $p < 0.001$, $V = 0.195$). The 30.1% of people in the "Inactive" group presented negative SPH, compared to 4.3% in the "Very active" group, $p < 0.05$. Conversely, 73.3% of people in the "Very active" group presented positive SPH, compared to 34.9% of people in the "Inactive" group, $p < 0.05$ (Table 2). Figures S1–S3 present the prevalences of positive SPH, depression and anxiety according to PAL.

Moreover, associations were found between mental health according to the GHQ-12 and PAL scores in the total sample and both sexes, $p < 0.001$. The highest medians GHQ-12 scores were presented by the "Inactive" group in the total sample (13.8 points), in women (14.9 points) and men (12.7 points), presenting significant differences with the rest of the

**Table 3** Associations between PAL and mental health score, according to Goldberg General Health Questionnaire GHQ-12 in Spanish adults with high cholesterol, SNHS 2017.

| | | | Overall | | | | |
|---|---|---|---|---|---|---|---|
| PAL | | GHQ-12 | PAL Comparison | M DIF. | m DIF. | K-W | M-W U |
| | Median (IQR) | 12.0 (8.0) | Inactive-Walker | 2.0 | 2.6 | | 0.011 |
| Inactive | Mean (SD) | 13.8 (6.9) | Inactive-Active | 3.0 | 3.9 | | <0.001 |
| | | | Inactive-Very active | 3.5 | 4.2 | | <0.001 |
| Walker | Median (IQR) | 10.0 (5.0) | Walker-Active | 1 | 1.3 | | <0.001 |
| | Mean (SD) | 11.2 (5.1) | Walker-Very active | 1.5 | 1.6 | <0.001 | <0.001 |
| Active | Median (IQR) | 9.0 (5.0) | Active-Very active | 0.5 | 0.3 | | 0.334 |
| | Mean (SD) | 9.9 (3.9) | | | | | |
| Very active | Median (IQR) | 8.5 (6.0) | | | | | |
| | Mean (SD) | 9.6 (3.9) | | | | | |
| | | | Women | | | | |
| PAL | | GHQ-12 | PAL Comparison | M DIF. | m DIF. | | |
| | Median (IQR) | 12.0 (9.0) | Inactive-Walker | 1.0 | 3.0 | | <0.001 |
| Inactive | Mean (SD) | 14.9 (6.7) | Inactive-Active | 2.0 | 4.6 | | <0.001 |
| | | | Inactive-Very active | 4.0 | 4.9 | | <0.001 |
| Walker | Median (IQR) | 11.0 (6.0) | Walker-Active | 1.0 | 1.6 | | <0.001 |
| | Mean (SD) | 11.9 (5.3) | Walker-Very active | 3.0 | 1.9 | <0.001 | <0.001 |
| Active | Median (IQR) | 10.0 (5.0) | Active-Very active | 2.0 | 0.3 | | 0.331 |
| | Mean (SD) | 10.3 (4.1) | | | | | |
| Very active | Median (IQR) | 8.0 (5.0) | | | | | |
| | Mean (SD) | 10.0 (4.4) | | | | | |
| | | | Men | | | | |
| PAL | | GHQ-12 | PAL Comparison | M DIF. | m DIF. | | |
| | Median (IQR) | 12.0 (6.8) | Inactive-Walker | 2.0 | 2.3 | | <0.001 |
| Inactive | Mean (SD) | 12.7 (6.8) | Inactive-Active | 3.0 | 3.2 | | <0.001 |
| | | | Inactive-Very active | 3.0 | 3.2 | | <0.001 |
| Walker | Median (IQR) | 10.0 (5.0) | Walker-Active | 1.0 | 0.9 | | 0.004 |
| | Mean (SD) | 10.4 (4.6) | Walker-Very active | 1.0 | 0.9 | <0.001 | 0.041 |
| Active | Median (IQR) | 9.0 (5.0) | Active-Very active | 0.0 | 0.0 | | 0.906 |
| | Mean (SD) | 9.5 (3.7) | | | | | |
| Very active | Median (IQR) | 9.0 (6.0) | | | | | |
| | Mean (SD) | 9.5 (3.6) | | | | | |

**Notes.**

IQR, Interquartile range; SD, Standard deviation; GHQ-12, Goldberg's General Health Questionnaire. Scores between 0 and 36. 0, the best mental health. 36, the worst mental health; PAL, Physical Activity Level: Inactive (PAI = 0; They declare not to go for a walk, no day a week, more than 10 min at a time). Walkers (PAI = 0; Report walking, at least one day a week, more than 10 min at a time). Actives (PAI = 1–30); Very actives (PAI = +30); PAI, Physical Activity Index: Scores between 0 and 67.5; K-W, $p$-value from Kruskal-Wallis test; M-W U, $p$-value from Mann Whitney $U$ test. M DIF (Median differences) and m DIF (Mean differences) in pairwise comparison of Goldberg's General Health Questionnaire between Physical activity groups.

PAL groups, $p < 0.001$. In contrast, the lowest medians scores were found in the "Very active" groups, in the total sample (9.6 points) and in women (10.0) and men (9.5 points). In none of these groups were significant differences found between "Active" and "Very active" groups (Table 3).

### Multiple binary logistic regression model for the risk factors for self-reported depression and anxiety

Being female, older, having lower social class and being physically inactive were risk factors for depression (Nagelkerke's $R^2 = 11.0\%$) according to the binary multiple logistic regression model for depression risk factors (Table 4). In addition, being female, older, of lower social class and physically inactive were also risk factors for self-reported anxiety (Nagelkerke's $R^2 = 7.5\%$) according to the binary multiple logistic regression model for anxiety risk factors. The same results were obtained in the models for mental health status (Nagelkerke's $R^2 = 8.8\%$) (Table 4).

## DISCUSSION

The main objective of this study was to analyse the dependence relationship between sex and SPH, anxiety and depression in Spanish adults with high cholesterol in the pre-pandemic period, as well as to assess the risk factors for negative SPH, depression, anxiety and mental health scores in this population, taking into account possible influencing factors such as different levels of physical activity, social class or sex.

### Main findings and theoretical applications
#### Depression, anxiety, self-reported health and mental health
Women with high cholesterol presented worse scores in psychological distress than men. Finding dependency relationships between the prevalence of depression, anxiety and self-reported health, mental health scores and sex. Specifically, women with high cholesterol had a higher prevalence of depression and anxiety than men. However, men had better self-perceived health. Based on these results, our study highlights the existence of mental health disorders such as depression and anxiety in people with high cholesterol. These results are in line with those reported by previous studies which have also associated depressive and anxiety states in people with high cholesterol (*Pereira, 2017*; *Troisi, 2009*; *Rafter, 2001*; *Cheon, 2023*). A possible hypothesis established by *Papakostas et al. (2003)* would be that an elevated cholesterol level would be responsible for decreased serotonin receptor sensitivity.

In contrast, some studies indicate that low cholesterol levels are associated with a greater presence of depressive symptoms, and even with greater cardiovascular events (*Steegmans et al., 2000*; *Barter et al., 2007*). In this regard, the study by *Han (2022)* observed a U-shaped relationship between cholesterol level and depressive symptoms in their study. In that study, low cholesterol levels <169 mm/dL and high cholesterol levels >222 mg/dL were a risk factor for depressive symptoms. In our study, we focused on specifically studying the high cholesterol population because of the clinical importance of addressing these individuals given the various health problems resulting from high cholesterol levels (*Sharrett et al., 2001*; *Curb et al., 2004*; *Tunstall-Pedoe et al., 1994*; *Al-Zahrani et al., 2021*; *Grau et al., 2007*).

On the other hand, in relation to the mental health assessed with the GHQ-12 in this population, median values of 10 were observed in the total sample, being worse in women 11 points than in men 10 points. These values would be below the established cut-off

Peer J

**Table 4  Logistic binary regression model for depression, anxiety and psychological distress (mental health) risk factor.**

| | Moder for depression | | | Moder for anxiety | | | Model for mental health | | |
|---|---|---|---|---|---|---|---|---|---|
| | Adjusted OR | (95% CI) | *p*-value | Adjusted OR | (95% CI) | *p*-value | Adjusted OR | (95% CI) | *p*-value |
| Age | 1.007 | (0.997; 1.017) | 0.173 | 0.989** | (0.979; 0.998) | 0.020 | 0.983* | (0.975; 0.991) | <0.001 |
| Sex | | | | | | | | | |
| Men | Reference | | | Reference | | | | | |
| Women | 2.577* | (2.110; 3.148) | <0.001 | 2.543* | (2.075; 3.116) | <0.001 | 1.733* | 1.461; 2.056 | <0.001 |
| PAL | | | | | | | | | |
| Very active | Reference | | | Reference | | | | | |
| Inactive | 2.144** | (1.350; 3.405) | 0.001 | 1.726** | (1.093; 2.725) | 0.019 | 2.737* | (1.855; 4.039) | <0.001 |
| Walker | 1.256 | (0.814; 1.939) | 0.303 | 1.126 | (0.737; 1.720) | 0.582 | 1.249 | (0.871; 1.792) | 0.226 |
| Active | 0.634 | (0.393; 1.024) | 0.063 | 0.846 | (0.537; 1.332) | 0.470 | 0.802 | (0.543; 1.184) | 0.267 |
| Social Class | | | | | | | | | |
| (I) | Reference | | | Reference | | | | | |
| (II) | 1.104 | (0.591; 2.061) | 0.757 | 0.842 | (0.465; 1.525) | 0.571 | 1.112 | (0.682; 1.814) | 0.620 |
| (III) | 2.189** | (1.369; 3.499) | 0.001 | 1.651** | (1.072; 2.543) | 0.023 | 1.734** | 1.186; 2.535) | 0.004 |
| (IV) | 1.934 | (1.183; 3.162) | 0.009 | 1.751** | (1.115; 2.749) | 0.015 | 2.129* | (1.440; 3.148) | <0.001 |
| (V) | 2.730* | (1.750; 4.257) | <0.001 | 1.186 | (1.242; 2.806) | 0.003 | 2.092* | (1.465; 2.987) | <0.001 |
| (VI) | 2.687* | (1.670; 4.323) | <0.001 | 2.100** | (1.351; 3.263) | 0.001 | 2.662* | (1.811; 3.914) | <0.001 |
| Constant | 0.032* | | <0.001 | 0.117* | | <0.001 | 0.264* | | <0.001 |

**Notes.**

OR, Odds ratio; CI, Confidence Interval.

*for *p*-value < 0.001.

**for *p* < 0.05.

point of 12 points, which would indicate no emotional disorders in total sample (*Muñoz Bermejo et al., 2020*). However, in our previous study also conducted in the pre-pandemic period in the general Spanish population, mental health scores were lower. A median of nine points was observed in the total sample and nine in men and 10 in women, which might suggest worse mental or emotional disorders in our sample of people with high cholesterol (*Denche-Zamorano et al., 2022a*).

### Physical activity levels, depression, anxiety and mental health scores and sex

The results show dependence relationships between the PAL and sex, with the proportion of men in "Very active" group being higher than that of women, although the number of women walkers was higher. Our results are consistent with previous studies, which showed that men are more active than women and tend to perform lower intensity activities (*Andruškiene et al., 2013*; *Denche-Zamorano et al., 2022c*; *Azevedo et al., 2007*). This could be related to our finding that women walk more than men. Similarity to our results, *Abel, Graf & Niemann (2001)* also report that women walked more than men in their study. In this sense, several studies show differences between sexes alluding to the participation of men and women in different areas of activity (activity at work or at home, for transportation and during leisure time) and at different intensities (moderate and vigorous) (*Guthold et al., 2018*). This difference in participation could explain our results.

It should be noted that the most relevant results of our study were observed when analysing the prevalence of depression, anxiety and mental health scores stratified according to the level of physical activity. In this analysis, a higher prevalence of depression was associated with the "Inactive" group. In addition, a lower prevalence was associate with people in "Active" group. In the case of anxiety, our results suggest that the highest prevalence was also found in "Inactive" group people and the lowest prevalence was observed in "Very active" group. In the same way, a 72.66% of physically very active people showed good self-perceived health. There is now strong evidence to support our findings that physical activity has a positive impact on self-perceived health, better mood control and anxiety status in people (*Denche-Zamorano et al., 2022c*; *Dostálová et al., 2021*; *Denche-Zamorano et al., 2022d*). In addition, the GHQ-12 mental health scores in the PAL-adjusted analysis, women and men in the "Very Active" and "Active" groups had lower scores than the "Inactive" group. The group of people with high cholesterol who were also physically inactive had median scores of 12 and mean scores of 13.8 on the GHQ-12. Physically inactive men and women had mean scores of 12.8 and 14.9, respectively. These values may imply emotional disturbances among physically inactive men and women with elevated cholesterol as they exceed the established cut-off point of 12 points for this questionnaire (*Muñoz Bermejo et al., 2020*). In contrast, the results obtained in the previous study were lower. Specifically, a median of 11 and a mean of 12.04 points in the physically inactive general population, and a mean of 11.57 points in men and 12.5 points in women in the "inactive" group on the mental health GHQ-12. Based on our results, it appears that physically inactive people with high cholesterol have worse psychological distress than the physically inactive general population, although studies comparing the

two populations would be needed. Furthermore, research supports the importance of addressing physical health complications (inactivity) in people with mental illness in order to improve their quality of life (*Rejeski & Mihalko, 2001*). Pointing that higher PAL protects people from depression (*Schuch et al., 2016*), anxiety, and other disorders (*McDowell et al., 2019*) compared to people with lower PAL (*Schuch et al., 2020*). Regarding the analysis by sex, it can be observed that among the physically inactive and very active women the scores decreased by an average of 4.9 points while among the physically inactive and very active men there was a decrease of 3.4 points on the mental health GHQ-12.

Our results may suggest that the level of physical activity could have a greater influence on women than on men. In contrast, *Li et al. (2022)* found no obvious difference in the role of PA in mental distress for men and women.

### Regression analysis

Additionally, to investigate factors influencing the prevalence of depression, anxiety and mental health in our sample of Spanish population with high cholesterol beyond physical activity levels, a regression analysis was performed. After adjusting the analyses for other factors in addition to PAL, such as age, sex and social class, being physically inactive was a significant risk factor for depression and anxiety and mental health symptoms. In a previous systematic review covering studies in populations with diverse characteristics, similar results have been reported (*Razzak, Harbi & Ahli, 2019*). In this review, female sex, family history of chronic illness, economic hardship/low socioeconomic status, stressful life events and lack of social support were found to be the most significant risk factors (*Razzak, Harbi & Ahli, 2019*). In addition, environmental factors such as the characteristics of dietary habits and other unhealthy behaviours have also been described as influencing one's SPH, reinforcing our assumption of social factors as a risk factor (*Andruškiene et al., 2013*; *Muñoz et al., 2008*). The authors consider that PA could be an adjuvant to pharmacological treatment that favours its results and limits undesired effects as well as the indiscriminate use of drugs (*Ivanovic & Tadic, 2015*; *Machón et al., 2016*; *Bonner et al., 2017*).

## Practical implications and future directions

Although it should be kept in mind that these are preliminary results and it is a cross-sectional study, these results establishing physical inactivity as a risk factor for depression, anxiety and mental health scores is a relevant finding, since the level of physical activity is a modifiable risk factor. Therefore, these results support the need to implement community health promotion programs through physical activity. from the authors' point of view, the introduction of physical exercise programs in this population could be a useful form of intervention to improve their mental health and help them meet physical activity recommendations (*Bull et al., 2020*; *ACSM's Guidelines for Exercise Testing and Prescription, 2021*).

On the other hand, one of the potentials of our results, in addition to the robustness of the results and the representativeness of the sample analysed, is that they are the latest data recorded in the pre-pandemic period in this Spanish population. This will allow us to conduct to carry out future studies to analyse the im-pact of the pandemic

on the relationships between the level of physical activity and mental disorders in this population. In addition, lines should compare the mental health of inactive people with high cholesterol with that of the general population, and possible interaction with sex, in order to see whether might benefit more, or what the optimal dose of physical activity for this type of population. Likewise, the development of longitudinal studies could establish the real impact of physical activity on people with high cholesterol.

### Limitations

This study contains some limitations. (1) inherent limitation to the cross-sectional study design so that causal relationships between variables cannot be established. (2) other variables that could influence depression and anxiety were not included, such as some sociodemographic, cultural, alcohol consumption or economic characteristics of the participants. It is advisable to develop future lines of research that address other behaviours such as nutritional or health habits such as taking medication, or the intake of substances such as alcohol as well as the prevalence of access to and use of health services and their characteristics.

## CONCLUSIONS

According to the results, women in this study had a higher prevalence of depression and anxiety than men. In addition, men were more likely to report being very active, although the proportion of walkers was higher in women. Likewise, our study found that PAL was associated with depression and anxiety in a sample of the Spanish resident population with elevated cholesterol levels. Higher proportions of depression and anxiety were found in both physically inactive men and women. Finally, being female, older, of low social class and inactive were factors in our sample for a higher probability of presenting depression and anxiety.

### Funding

The APC was funded by Open Access Program of Universidad de Las Américas. Ángel Denche-Zamorano (FPU20/04201) was supported by a grant from the Spanish Ministry of Education, Culture and Sport, Grants FPU20/04201, funded by MCIN/AEI/10.13039/501100011033 and by "European Social Fund Investing in your future" associated with the "European Union NextGenerationEU/PRTR". Maria Mendoza-Muñoz was supported by a grant from the Universities Ministry and the European Union (NextGenerationUE) (MS-12). Damian Pereira-Payo was supported by a grant from "Plan Propio de Iniciación a la Investigación, Desarrollo Tecnológico e Innovación de la Universidad de Extremadura". Raquel Pastor-Cisneros was supported by a grant from the Valhondo Calaff Foundation (Cáceres, Spain). The funders had no role in study design, data collection and analysis, decision to publish, or preparation of the manuscript.

### Grant Disclosures

The following grant information was disclosed by the authors:

Open Access Program of Universidad de Las Américas.

Spanish Ministry of Education, Culture and Sport: FPU20/04201.

"European Social Fund Investing in your future" associated with the "European Union NextGenerationEU/PRTR": MCIN/AEI/10.13039/501100011033.

Universities Ministry and the European Union (NextGenerationUE) (MS-12).

"Plan Propio de Iniciación a la Investigación, Desarrollo Tecnológico e Innovación de la Universidad de Extremadura".

Valhondo Calaff Foundation (Cáceres, Spain).

## Competing Interests

The authors declare there are no competing interests.

## Author Contributions

- Ángel Denche-Zamorano performed the experiments, analyzed the data, authored or reviewed drafts of the article, and approved the final draft.
- Jofre Pisà-Canyelles conceived and designed the experiments, analyzed the data, authored or reviewed drafts of the article, and approved the final draft.
- Sabina Barrios-Fernández performed the experiments, prepared figures and/or tables, and approved the final draft.
- Antonio Castillo-Paredes performed the experiments, analyzed the data, authored or reviewed drafts of the article, and approved the final draft.
- Raquel Pastor-Cisneros performed the experiments, analyzed the data, prepared figures and/or tables, and approved the final draft.
- Maria Mendoza-Muñoz performed the experiments, prepared figures and/or tables, and approved the final draft.
- Diana Salas Gómez analyzed the data, prepared figures and/or tables, and approved the final draft.
- Cristina Mendoza Holgado conceived and designed the experiments, authored or reviewed drafts of the article, and approved the final draft.

## Data Availability

The raw data are available in the Supplemental Files.

## Supplemental Information

Supplemental information for this article can be found online at http://dx.doi.org/10.7717/peerj.17169#supplemental-information.

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
