# Peer review of "Evaluation of the association of physical activity levels with self-perceived health, depression, and anxiety in Spanish individuals with high cholesterol levels: a retrospective cross-sectional study"

_PeerJ, doi:10.7717/peerj.17169_

## Round 0.1 · original submission · Major Revisions

Dear authors,

In addition to the reviewers' suggestions, I have some comments on your article.

Depression and chronic anxiety status: The Spanish National Health Survey includes 3 different variables to describe health situations, a, b and c, being b ”Have you suffered this health situation during the last 12 months?” and c ”Has any health staff diagnosed you this health situation?”. The correct questions are G25a_20 and G25a_21. Moreover, if the authors want to describe accurately the presence of depression and chronic anxiety they perhaps should use the questions G25b_20, G25c_20, G25b_21, and G25c_21 considering that the information is available. The same consideration is valid for the Cholesterol status.

Table 2. Please review the footnote. It should include the meaning of a, b, and c.

Tables 4, 5, and 6. It is not suitable to include the tables in the very same format of SPSS, as they have some non-relevant information; another more simple format could be better. The term Logarithm binaric regression is not adequate.

Considering all these issues I must recommend that you make major changes in order to reconsider the decision.

Hoping for your new version. Thanks and best wishes.

Reviewer 1 ·

Basic reporting

Im reviewing this manuscript for the first time and I don’t have much experience with analyses of survey data. I found the paper somewhat of a challenge, mostly do with framing of results and format of tables and figures (or lack thereof).

1. The theoretical framing of study is mostly ok, though the section on relationship of lipid traits on mental health could be rewritten for clarity. To emphasize, what are associations, what are causations, and what is unknown? For instance, is there any basis for the sexual dimorphism hypotheses posed there?
2. The results are presented as series of tables, stating mean differences between groups/conditions. Later are results from multifactor linear models presented. I would have liked the linear models come first, because they will inform the reader of the impact of each variable, before the breakdown of what increases/decreases in which subgroup is presented. Also, the research questions are not clearly stated in intro, and not refreshed in the results section. Like “We set out to study if A and B (or their interaction) influenced C and D.” Also, there is no attempt to test for interaction of variables, which is actually one of the main conclusions of the paper (Line 356 “being female, older, of low social class and inactive were factors in our sample for a higher probability of presenting depression and anxiety”).
3. The text needs a read-over for improving the language – several suggestions are provided as “minor points” but those are far from exhaustive. “In this regard”, “in this sense” are used repeatedly in the text, and in some case wrongly. When simpler “bridging words” would suffice.

Experimental design

4. Uncertainty about the sources data. Did you do the survey or did you used data from the survey? Line 110-223 “For this purpose, the adult questionnaire was used…” If you did the survey for this study, then provide all details. If not, just emphasize the main aspects that matter to explain the nature of your dataset, its strengths and limitations. And cite relevant source.

Validity of the findings

5. Crucially, there is no description of how the linear models where evaluated. How well do the assumptions of the diff. tests hold, how are residuals distributed etc? Line 191-208.
6. Several figures and table where challenging to understand. See list below.

Table 1 is quite confusing. In particular the different statistics columns, that apply to some rows but not others. Also, for the SPH rows and below I think the numbers refer to means or SD, but it is not indicated. If SD then it should be in brackets as the rows above. Hard to know what the CC refers to, CC for which aspect of the data?? Perhaps just best to split table in two based on nature of data/statistic?

Figure 1. Several issues. ADD A, B, C to each panel. The X axis title “sex” is in line with the catgories “male”,”female”. The legend should indicate that the Y axis is broken for panel A. Specify the age limits of the cohort, and indicate it nationality. Don’t see need for the same inset in all three panels to indicate shade of grey for sex. Same applies to Y label.

Several issues with Table 2. I suggest the authors make a figure version (A, B, C) of table 2. Should keep the table (in supplement maybe). The a,b,c to indicate significance render the table hard to read. And is also not indicated clearly! Format of table is confusing also, its hard to know what the Yes/no applies to, as the “depression” etc is centered in the row. % is explained in footnote, but it is unclear which column has percentages.

Table 3. Again, madly confusing representation. What comparsions are the M dif and m dif representing? Difference of Inactive Walker to what?? Legend mostly understandable, but rest is very confusing.

Table 4-5
“Wald (Wald Chi-Squared Test)” – the test statistic, not the test.

Table 6.
No significance is 0.
St. = standard deviation??

Additional comments

Minor points.

Abstract: Add “Conclusion” subsection. The last sentence of the abstract is a conclusion.

Line 21.
Reword “It presents a high”, “It has a high …” and skip “being”.
Line 26.
Reword “Globally, the adult suffers”, “Globally, adults suffer…”
Line 29.
Reword “that raise the imperative need to “
Line 30.
Reword “Within the prevention…will” , “Among the prevention….could ” (and skip the to)
AND add citation or mentioning of the “prevention” strategies.
Line 39.
Add estimates of magnitude on the cholesterol to mental health risk (1-10-25%?)
Line 53.
Not the right connection words “In this regard,” find replacement
Line 59
“self-rated health status is a factor associated” – is the word “factor” needed in this sentence?
Line 67
Is lifestyle a single factor?
Line 75-6.
Covid19 repeated in the sentence – and same reference cited twice in same line. And also, is this your own study? Perhaps indicate that.
“PAL in the pre-pandemic period analysing the data of the Spanish National Health Survey (SNHS 2017) [31], the last one before the COVID-19 pandemic [31].”
Line 78-
You talk of authors. Yourselves or the authors of the previous study. Same people?
Line 80
“can be” TO “could be”

Line 83.
Reword “According to the above, the” Maybe “According to the literature cited above, “
And, clarify correlation does not mean causation. Do you actually expect bidirectional effects? Or do you mean they could be in one direction or the other?

Line 86-7.
Reword “similar or no in men and women”

Line 89.
This sentence sticks out – perhaps just include in the discussion “In addition, the results will provide baseline data on mental health in this population before the COVID-19 pandemic”
Or explain at some point that you will use data from year x to y, and that this is a pre covid situation.

Line 93
Skip “Therefore”
Consider framing the aims as RQ´s or hypotheses.

Line 148
What does “NS/NC” mean?

Line 158.
Justify the 12 as cutoff, and consider rewording “previous study presented the psychometric properties of this questionnaire”

Line 181.
Mind the underline “Sex:”

Line 213.
Suggest reiterate the filtering of samples for the dataset. Starting with “Table 1 shows” reads a bit like a bachelor project paper.

Reviewer 2 ·

Basic reporting

no comment

Experimental design

no comment

Validity of the findings

no comment

Additional comments

no comment

Reviewer 3 ·

Basic reporting

Basic reporting is done better than previous submission of this article.

Experimental design

The question and purpose has been defined well. However the established research purpose of prevalence and factors of mental health issues among high cholesterol level individual based on their physical activity level is a commendable choice but the undertaken analysis and approach doesn't sufficiently address the issue.

Validity of the findings

The authors have failed to establish why the choice of certain demography was made. The entire research basically finds out that if the PAL is low (inactive) then there are high prevalence and odds for depression, anxiety and mental health (GHQ-12) among high cholesterol individuals and the odds were also higher for females, elders, and people of low social class. Fair enough. But my question to the authors is, how is that specific to high cholesterol individuals? As far as my understanding goes about relationship between mental health and PAL, the same result (that inactive individual will have higher odds of mental health issues) would be obtained for any demography be it low cholesterol individual or general population. Multiple such studies have already established, as reported in the Literature review of your paper, that PAL can effect the mental health level and SPH of various demographic. If any comparison between other demography such as low cholesterol individual or general population and high cholesterol individual have been made, then a different and more clear picture of the issue would have been presented itself. Any specific finding related to high cholesterol individual was not made in this research article. Hence, it does not not add to the understanding of mental health issues among high cholesterol individuals and is not publishable.
Secondly, a difference in the prevalence of depression, anxiety and mental health based on the PAL was obtained among males and females. However, authors have not explained or explored the possible reasons for such variation in the Discussion and brushed it off.

Reviewer 4 ·

Basic reporting

1) "The inactive population...........................................................SPH, depression, and anxiety", written repeatedly in the abstract.
2) The sentence in line no. 46-49 seems lengthy; rewrite it to increase its readability.
3) In line no. 62, "because it's are important............", Check the grammar.
4) In line no. 87, you should write "not" instead of "no".
5) In line no. 95, you have written 'SHP', which shows carelessness; I hope there is no abbreviation like this.
6) The 'Study design and sample' section seems lengthy. Rewrite it to make it more readable, and don't write too much. Things are also mentioned repeatedly.
7) In table no. 2, alongside the values, you have mentioned the letters 'a', 'b', and 'c'. Explain them.
8) In the 'results' section (line no. 238), you described the medians, but those values in table no. 3 are of means. If you are describing one value and can not take care of both 'mean' and 'median' values, you should put the one in the table.
9) You can make a single table for the results of risk factors of depression, anxiety, and mental health rather than making it separate. It will make your article more readable.
10) Don't put results table by table. Also rather than describing the results, you should explain the results. While explaining the results, just mention table no. where you have mentioned the data from that table.
11) In line no. 258, you mentioned, "among others". what does it mean? is there any other variable than age?
12) In line no. 291. what is the reference for 'the fact' that women have a greater affinity?
13) Discussion seems a little weak. as the findings seem to be very general.

Experimental design

In the Introduction part, authors mentioned, "to the authors’ knowledge, there were no data on the association between mental disorders such as depression and anxiety and PAL in a specific Spanish sample such as the high cholesterol population also in the pre-pandemic period. Then, the authors consider that improving physical fitness can be a strategy to address the impact of an unhealthy lifestyle on depression and anxiety in Spanish people with high cholesterol".
So my query is if, on a specific group/ demography (here, high cholesterol), there is no data on establishing the relation between depression, anxiety, and PAL. Would you create data on every demographic separately? Would you like to create data on people with Low cholesterol levels also?
Have you checked whether the findings are similar or different with low-cholesterol people, before selecting a high-cholesterol demographic group?

Validity of the findings

I think the findings would be the same for general people and people with low cholesterol levels, so there is no reason for selecting a specific demography. Have the authors checked it or jumped directly to the specific group?
With all these comments I would say the study lacks novelty.

Additional comments

No comments

Reviewer 5 ·

Basic reporting

Line 14 onwards: Instead of using “active” or “inactive,” the authors can use “physically inactive” or “physically active.”
Line 93: No need to use “similar or no” when “whether or not” is already mentioned in the same sentence (line 92).
Lines 100-103: The first and second objectives seem similar, so they can be merged into a single objective.
Lines 138-140: The figure explains what is written in the text; therefore, it is unnecessary to include redundant information. Simply mention that a detailed sample selection procedure is explained in Figure 1. It would be better to refer to the “questions asked” rather than “question numbers” in the manuscript for better reader comprehension. Please refer to this paper to understand what I want to convey. https://doi.org/10.1038/s41598-023-44762-8
Instead of dividing Procedures and Variables into multiple subsections, it is recommended to keep only two subsections: one for “outcome variables” and another for “explanatory variables.” Socio-demographic characteristics should also be included in the “explanatory variables” section.
Cholesterol status is mentioned as an explanatory variable in subsection 2.2.2 but is not present in the regression table. If it is not included, there is no need to mention it as an explanatory variable.
Table 2: Rows and columns can be transposed for better clarity.
The tables should be modified before including it into the manuscript. It should not include each and every part of output which software gives. In logistic regression table only exponential beta (also known as odds ratio), its 95% confidence interval, and significance level are most crucial information and only they should be included in the table. Tables 4 and 5 can be merged into a single table, with only crucial information included. A single table with five columns should be created: the first column showing predictors (e.g., age, sex, physical activity level, social class), the second and third columns for odds ratios and their 95% confidence intervals for depression, and the fourth and fifth columns for odds ratios and their 95% CIs for anxiety. Significance levels can be denoted using stars in the superscript of odds ratios (e.g., *** for p-value < 0.001, ** for p < 0.05, and * for p < 0.10).
Line 318: There is a spelling mistake; it should be “scores,” not “cores.”

Experimental design

No comment

Validity of the findings

No comment

---

## Round 0.2 · accepted · Accept

The authors have addressed in a suitable way all of the reviewers' and editor's comments. I have assessed the revision and I think that the manuscript is ready for publication.